# Biofortification and Valorization of Celery byproducts Using Selenium and PGPB under Reduced Nitrogen Regimes

**DOI:** 10.3390/foods13101437

**Published:** 2024-05-07

**Authors:** Jacinta Collado-González, María Carmen Piñero, Ginés Otálora Alcón, Josefa López-Marín, Francisco M. del Amor

**Affiliations:** Department of Crop Production and Agri-Technology, Murcia Institute of Agri-Food Research and Development (IMIDA), C/Mayor s/n, 30150 Murcia, Spain; mariac.pinero2@carm.es (M.C.P.); gines.otalora@carm.es (G.O.A.); josefa.lopez38@carm.es (J.L.-M.); franciscom.delamor@carm.es (F.M.d.A.)

**Keywords:** celery, nutritional quality, selenium, nitrogen dose, plant growth-promoting bacteria, valorization

## Abstract

Due to climate change and exacerbated population growth, the search for new sustainable strategies that allow for greater food productivity and that provide greater nutritional quality has become imperative. One strategy for addressing this problem is the combined use of fertilization with a reduced dose of nitrogen and biostimulants. Celery processing produces a large amount of waste with its concomitant pollution. Therefore, it is necessary to address the valorization of its byproducts. Our results revealed reductions in the biomass, Na, P, Mn, B, sugars, and proteins in the byproducts and increased lipid peroxidation, Fe (all celery parts), and K (byproducts) when the N supplied was reduced. Plants inoculated with *Azotobacter salinestris* obtained a greater biomass, a higher accumulation of K (byproducts), a build-up of sugars and proteins, reduced concentrations of P, Cu, Mn, B, Fe (petioles), and Zn (byproducts), and reduced lipid peroxidation. The application of Se at 8 μM reinforced the beneficial effect obtained after inoculation with *Azotobacter salinestris*. In accordance with our results, edible celery parts are recommended as an essential ingredient in the daily diet. Furthermore, the valorization of celery byproducts with health-promoting purposes should be considered.

## 1. Introduction

Celery is one of the most important plants in the Apiaceae family. It is an excellent source of proteins, vitamins, antioxidant compounds, and minerals beneficial to human health [1]. It is very rich in bioactive compounds and, due to its aroma and flavor, is widely consumed worldwide either raw in salads or cooked as a condiment [2]. The broad biological activity of all the bioactive compounds present in celery grant it with the possibility of preventing and treating several diseases, including gastrointestinal disorders, cardiovascular and liver diseases, asthma, male fertility problems, and cancer [3]. Nonetheless, during celery processing, as with other vegetables, a large volume of waste is generated due to the removal of the petioles and external leaves, which are not usually consumed. This practice has substantial environmental and economic effects [2]. For this reason, ways to revitalize these byproducts are currently being sought so that their processing can be reduced and to contribute to the circular economy [4].

It is well known that nitrogen (N) plays an essential role in the growth and development of plants [5]. Thus, the appropriate provision of N to crops results in greater productivity and a greater accumulation of phytochemicals [6]. For this reason, nitrogen fertilizers have been and are widely used in agriculture. However, scientific studies have reported that an excessive supply of N in the form of nitrates has detrimental effects not only on plants but also on the environment and human health [7,8]. One of the main concerns is the relationship between the intake of nitrates in the diet and the incidence of suffering from methemoglobinemia, which causes blue baby syndrome. This disease is common in babies who have ingested a high concentration of nitrates through food. Another important worry is the conversion of nitrates into highly carcinogenic compounds called nitrosamines [7]. Consequently, in recent decades, the application of nitrogen fertilizers to crops has decreased [9].

An alternative option to using large amounts of traditional nitrogen fertilizers may be the use of plant growth-promoting bacteria (PGPB). Among the PGPB most used as biofertilizers, we find bacteria from the genus Azotobacter. Previous studies have shown that plants that were inoculated with Azotobacter bacteria and fertilized with a reduced N supply had a similar productivity to plants fertilized with nitrogen fertilizers [8]. Furthermore, under these conditions, the plants were also found to have greater accumulations of bioactive compounds [7,10].

Selenium (Se) is an essential mineral in human and animal nutrition. It plays a unique role among all the essential trace elements, being necessary for the synthesis of proteins [11]. However, Se in excessive amounts (400 µg of Se per day) can be toxic, although a prolonged deficiency in Se in the human body causes serious diseases [12]. For this reason, the opinion on this mineral has shifted from it being considered toxic to it being attributed to notable health benefits, as it is important for male reproductive biology and the function of the thyroid gland, muscles, and cardiovascular, immunological, and central nervous systems [13]. Currently, about 1 billion people suffer from Se deficiency due to the low dietary intake of this mineral [12]. The Se contents in foods largely depend on the concentrations of this mineral in the soil. It is possible that this worrying scenario will worsen in the future as a consequence of climate change because, according to experts, climate change could promote Se deficiency in the soil, mainly in agricultural areas [13]. Studies have found that Se not only stimulates the growth and development of plants but also provides them with a greater tolerance to various types of abiotic stresses [14]. Furthermore, Cunha et al. [15] reported that the application of Se at low concentrations could increase the accumulations of bioactive compounds in soybean plants.

Based on the above, the aim of the present work was to study how inoculation with the PGPB *Azotobacter salinestris*, both in isolation and in combination with the application of Se, affects the biomass, accumulation of sugars and proteins, mineral content, and lipid peroxidation in different celery plant parts (petioles, leaves, and byproducts) under different reduced N regimes. This work also provides new information on this new, sustainable strategy, which includes the combination of an adequate concentration of Se, inoculation with *Azotobacter salinestris*, and the fertilization of plants with an appropriate dose of N. With the new strategy proposed in this work, the aim is to enhance the possible valorization of celery byproducts for purposes beneficial to health in the cosmetic and nutraceutical sectors.

## 2. Materials and Methods

### 2.1. Experimental Design and Treatments

The experiments were carried out on celery (*Apium graveolens* L.) cv. Gladiator (Babyplant S.L., Santomera, Murcia, Spain), randomly set up and grown in a polycarbonate greenhouse with two identical modules located in Murcia, Spain (37°56′27.3″ N, 1°08′01.8″ W). Seventy-two seedlings with nine true leaves were transplanted in plastic bags filled with coconut coir fiber (Pelemix, Alhama de Murcia, Murcia, Spain) at a spacing of 33 cm along the rows, which were 1 m apart. Celery plants were grown under controlled climatic conditions: the temperature (°C) and relative humidity values were 28/15 °C (day/night) and 70%. Three fertilizer treatments were used as the irrigation treatments for 72 plants (24 per treatment). The control treatment (100% N supply) was irrigated with a Hoagland’s nutrient solution composed of Ca(NO_3_)_2_·4H_2_O (362.0 mg L^−1^), KNO_3_ (404.4 mg L^−1^), K_2_SO_4_ (131.1 mg L^−1^), MgSO_4_ ·7H_2_O (123.2 mg L^−1^), H_3_PO_4_ (0.101 mL), and micronutrients. Treatment 2 had a mild N supply deficiency (60% N supply) and treatment 3 had a severe N supply deficiency (30% N supply). After transplanting, half of the plants from each irrigation treatment (twelve per treatment) were inoculated with the *Azotobacter salinestris* strain CECT9690, a PGPB, obtained from Ceres Biotics Tech, S.L. (Madrid, Spain). According to the manufacturer’s recommendation, 10 mL of inoculant per plant at a concentration of 250 µg mL^−1^ was applied individually to each pot and mixed homogenously with the substrate. After that, a total of 12 plants per nutrition treatment (6 inoculated and 6 non-inoculated) were randomly sprayed with Se as sodium selenate [Na_2_SeO_4_] at a concentration of 8 μM every 15 days. In a previous work by the authors of the present study, it was observed that a concentration of 8 μM provided the best beneficial results for both the health of the plant and its health-promoting effects [16]. In this way, the experimental design was a completely randomized design in a factorial arrangement, with six replicates (*n* = 6) and seven treatments. After 88 days, all the plants were harvested and separated into 2 parts: edible (central) and non-edible (outer) parts. In turn, the edible parts were divided into petioles and leaves, while the outer parts (both leaves and petioles) were considered as byproducts. In addition, the petioles, leaves, and byproducts were divided into two parts: one part was frozen at −80 °C until the lipid peroxidation analysis, and the second part was freeze-dried for the subsequent analysis of the sugars, total protein content, and cations.

### 2.2. Plant Biomass Measurements

On the last day of the experiment, before harvesting the plants, the heights of the celery plants were determined. After the harvest, the fresh weights and diameters of the celery plants were measured in the same plants from the treatment pots. After weighing the shoots of the intact plants, the celeries were divided into edible parts, composed of petioles and leaves (internal part), and non-edible parts, composed of leaves and petioles from the outer parts (byproducts). The edible parts were weighed and separated into petioles and leaves. Also, these celery parts, together with the byproducts, were used for the different analytical determinations. The dry weights of the different plant parts were estimated after freeze drying for 3 days, and the samples were weighed again to determine the dry shoot weights (DWs).

### 2.3. Chemicals and Reagents

Authentic standards for glucose, sucrose, fructose, inositol, thiobarbituric acid (TBA), trichloroacetic acid (TCA), 3,5-Di-tert-4-butylhydroxytoluene (BHT), and sodium selenate were purchased from Sigma-Aldrich (Steinheim, Germany), and absolute ethanol was acquired from Panreac Química (Barcelona, Spain). SPE cartridges (C18 Sep-Pak cartridges) were purchased from Waters Associates (Milford, MA, USA), and the ultrapure water used was obtained from a Millipore water purification system.

### 2.4. Determination of Cations

For the determination of the cations (Ca, K, Mg, B, Cu, Fe, Mn, P, and Zn) present in the celery leaves, petioles, and byproducts, ground, freeze-dried samples (0.1 g) were microwave acid-digested by adding 8 mL of concentrated nitric acid: water (1:1 *v*/*v*) and 2 mL of hydrogen peroxide in an ETHOS ONE microwave digestion system (Milestone Inc., Shelton, CT, USA) for 1 h. The digests were diluted to 25 mL with Milli-Q water. The cation contents in the digested samples were analyzed with an inductively coupled plasma (ICP) spectrometer (Varian Vista MPX, Palo Alto, CA, USA). The cation contents of the macronutrients and micronutrients were expressed as g kg^−1^ DW and mg kg^−1^ DW, respectively.

### 2.5. Determination of Sugar Contents

The amount of free soluble sugars was measured using the Balibrea et al. [17] method. For this purpose, a methanolic extract was prepared twice from 50 mg of freeze-dried celery tissues at 4 °C, for 30 min each time. Subsequently, each methanolic extract was then centrifuged at 3500× *g* for 15 min, at 4 °C. Then, the supernatants were filtered through a C18 Sep-Pak cartridge (Waters Associates, Milford, MA, USA) and combined and re-filtered through a 0.45 μm filter (Millipore, Bedford, MA, USA). The soluble sugars (20 μL) were analyzed with ion chromatography (METRHOM 861 Advanced Compact IC; METROHM 838 Advanced Sampler) (Metrohm, Herisau, Switzerland). Authentic standards were used as the calibration curve. Total and individual free soluble sugars were expressed as g kg^−1^ DW.

### 2.6. Determination of Protein Content

The total protein contents were analyzed in freeze-dried celery tissues (after at least 72 h at 65 °C) using a combustion nitrogen/protein determinator (LECO FP-528, Leco Corporation, St. Joseph, MI, USA). The crude protein values were calculated as the mineral nitrogen multiplied by the protein factor (6.25). The results were expressed as g 100 g^−1^ DW.

### 2.7. Determination of Lipid Peroxidation

The lipid peroxidation in the celery samples (leaves, petioles, and byproducts), in terms of thiobarbituric acid-reactive substances (TBARSs), was measured with a thiobarbituric acid (TBA) reaction. For this purpose, 3 mL of trichloroacetic acid (TCA) (20% (*w*/*v*)) was added to 100 mg of freeze-dried celery tissues. The mixture was homogenized and centrifuged at 3500× *g* for 20 min. Then, 1 mL of TCA (20%, *w*/*v*) containing TBA (0.5%, *w*/*v*) and 150 µL of BHT (4%, *w*/*v*) in ethanol was added to 1 mL of supernatant and mixed. The homogenate was incubated for 30 min at 95 °C and then cooled on ice and centrifuged at 10,000× *g* for 15 min. The absorbance of the resulting supernatant was measured at 532 nm using a UV–visible spectrophotometer (Shimadzu UV-1800 model with the CPS-240 cell holder, Shimadzu Europa GmbH, Duisburg, Germany). In addition, the value for the non-specific absorption at 600 nm was recorded. An extinction coefficient of 155 mM^−1^ cm^−1^ was used for calculating the concentration of thiobarbituric acid-reactive substances (TBARSs) [18]. Results were expressed as TBARSs µmol g^−1^ FW.

### 2.8. Statistical Analysis

The experiment was designed based on a completely randomized factorial experiment (3 × 2 × 2), and all of the data were analyzed with SPSS software v.25 (IBM, Chicago, IL, USA). All data were verified for homogeneity of variance and normality of distribution and were subjected to an analysis of variance (ANOVA) at a 95% confidence level. The factorial scheme corresponded to three N doses supplied (100%, 60%, and 30%), two treatments with PGPB (inoculated and non-inoculated plants), and the supply/absence of Se. The analyses were performed using five repetitions per treatment.

## 3. Results and Discussion

### 3.1. Determination of Celery Biomass

The data presented in Figure 1 show that the average weights of the total plant and edible parts were 2017 g and 861 g, respectively, the height was 57 cm, and the diameter was about 6 cm. In addition, the plants presented 9%, 18%, and 16% of dry weight in the petioles, leaves, and byproducts, respectively (Figure 2). The heights and petiole diameters were similar to those found for the dulee, rapeceum, and self-whitening Golden Clause cultivars [19,20]. The weights of the edible parts were similar to those of the Samurai and Primus celery varieties but lower than those of the Elixir and Atlant ones [3].

Our results revealed that the heights, diameters, fresh yields, and dry matter of the celery plants were affected by all the factors studied in this work: the percentage of N applied, the presence of PGPB, and the application of Se at 8 µM (Figure 1).

The reduction in the percentage of N applied in the nutrition solution had a substantial effect on the celery plant biomass, including the height and petiole diameter (Figure 1). Thus, the growth parameters were greatly reduced as the N supply was reduced. In this sense, the height was reduced by 22%, the edible-part and total plant weights were reduced by 32% and 33%, respectively, at the 60% N supply, and by 37% and 46%, respectively, at the 30% N supply, in non-inoculated plants and in the absence of the treatment of Se. The diameter showed a higher reduction when the 60% N supply was considered (40%) in relation to the control sample (100% N supply).

The dry matter was also negatively affected by the N starvation, and considerable reductions of 12% and 24% in the leaves and byproducts, respectively, were observed with the 30% N supply, in relation to the control sample (100% N supply) (Figure 2). These results are in accordance with previous results found in sorghum, corn, olive, and lettuce, which confirmed the positive relationship between the N supply level and the accumulation of biomass [21,22].

The reduction caused by the deficiency in N in the nutritive solution observed in Figure 1 and Figure 2 was reversed by the action of the PGPB. In fact, the heights increased by 15% as a result of the PGPB application, as compared to those plants that were only under N starvation. Several studies have indicated that the growth and development of plants and crop yields can be enhanced in plants inoculated with PGPB. It is known that PGPB have a direct effect on plant growth through several mechanisms, such as nitrogen fixation and the secretion of biologically active hormones, which are key to the growth and development of plants [8].

As can be observed in Figure 1, the Se treatment increased the accumulation of biomass. In this sense, an increase of 18% in the diameter and 21% in the edible-part weight with respect to the plants affected only by N deficiency can be observed. Ashraf Ganjouii et al. [14] reported that, at a low concentration, Se can ameliorate the detrimental effects of many abiotic stresses on the plant growth and yield by increasing the function of antioxidant enzymes. In this way, Se has the ability to increase the plant’s resilience.

### 3.2. Determination of Cations

Table 1 and Table 2 show the contents of macroelements and microelements present in three different celery parts. The most accumulated mineral elements in the celery were potassium (K), calcium (Ca), iron (Fe), and manganese (Mn). The highest mineral content was noted in the byproducts, followed by the leaves. It is important to note that the byproducts were also composed of leaves. In the current work, it was found that celery cv Gladiator contained higher K, Mn, and Zn contents and lower Fe contents, both in the petioles and leaves, than other cultivars, such as Elixir, Samurai, Atlant, and Primus. Celery cv Gladiator also contained leaves poorer in Cu than the Samurai, Atlant, and Primus varieties, and petioles richer than those of the cited varieties [3]. It is important to emphasize that the mineral distribution between the different celery tissues and even between different cultivars is dependent on the celery variety and fertilizer supplied [23]. Some of the elements studied (Ca, K, Zn, and Se) are involved in the antioxidant system of the plant [3,24]. Taking this into account, the mineral composition presented by each of the studied celery parts can provide valuable information on the contribution of nutrients obtained through fertilization, the resilience of the plant against abiotic and biotic stresses, and the nutritional quality of each of the parts. Our results show that there are significant differences between the different celery parts and the data obtained, depending on the N supplied in the fertilizer solution, the inoculation with PGPB, and the application of Se at 8 µm. In this sense, by reducing the N% input, significant reductions in Na, P, Mn, and B and considerable increases in Fe and K were observed, mainly in the byproducts (Table 1 and Table 2). Similar results were obtained for lettuce by other researchers [8,25].

The efficiency of the inoculation with *Azotobacter salinestris* increased when the N supply was lower than 100% (control), leading to the absorption and accumulation of different macro- and micronutrients in the different parts of the plant. The inoculation of the plants with PGPB resulted in lower contents of Cu, Mn, and B. Likewise, the application of PGPB also led to lower accumulations of Fe in the petioles and of Zn in the byproducts and to greater accumulations of K and Mg in the byproducts (Table 1 and Table 2). These results are in accordance with those found in a study on lettuce by Ikiz et al. [25], in which the authors found that the abundance and efficiency of PGPB can be affected by the applied fertilizer regimes. In fact, these authors found that the inoculation of PGPB with reduced N fertilization can lead to plants with greater nutrient absorption. This may indicate plants with an optimized distribution of available nutrients. The increases in the K and Mg contents are coherent with the results of Consentino et al. [8], who showed that PGPB were able to stimulate their accumulation in plant tissues. Similar to Hungria et al. [26], the P content was not affected by the inoculation with PGPB.

The application of Se resulted in plants with lower contents of Na, Ca, Mg, Fe, and Cu and greater accumulations of K, P, Mn, and Zn (Table 2). These results agree with those found for mature lettuce, coriander, and basil plants, in which the K, P, Ca, and Mg concentrations increased when Se was applied to the plants, whereas our outcomes were contrary to what was found for tatsoi [27,28]. Furthermore, other studies showed that the concentrations of Fe, Zn, and Mn increased in tatsoi microgreens, black-grained wheat, and rice but decreased in lettuce and white clover as a consequence of the application of Se to these plants [29,30,31]. According to all of these conflicting findings, it can be concluded that the concentrations of macronutrients and micronutrients may increase or decrease depending on the species and/or cultivar [27]. There is evidence that a high sodium intake is related to an increased risk of certain chronic diseases, including heart disease in humans. For this reason, the maximum recommended dose of sodium to be ingested in the diet of the adult population is 2 g/day, and it is 1.1 g/day in the diet of infants under 3 years of age [32].

Our results show that the fertilization of celery cv Gladiator with a reduced dose of N, together with the inoculation of bacteria and the application of Se at a concentration of 8 µm, satisfactorily reduced the concentrations of Na accumulated in different parts of the celery plant. Thus, 200 g of petioles, leaves, and byproducts provide only 0.071 g, 0.090 g, and 0.087 g of Na, respectively. These data are equivalent to 3.55%, 4.5%, and 4.3% of the RDA for adults, and to 6.45%, 8.18%, and 7.82% of the RDA for infants.

There is a close relationship between sodium and potassium. While sodium can increase blood pressure, potassium helps reduce it. Therefore, in recent years, due to the high incidence of people with cardiovascular problems, experts have been recommending that potassium intake be increased in diets [33]. According to the EFSA [34], the adequate daily intake for an adult, regardless of gender, should be 3500 mg/day. In our study, 200 g of celery petioles and leaves of the cv Gladiator grown under a reduced N regime, after the inoculation of bacteria and the application of Se, contained 0.95 and 1.4 g of K, which corresponds to 31.7% and 46.7%, respectively, of the recommended daily consumption. These values are exceeded by the K provided by the byproducts (52% of the recommended daily consumption).

Fe and Zn play a crucial role in the development and functioning of the human cognitive system. Currently, Fe and Zn deficiencies are very common in many populations around the world. In fact, 2 billion people are affected by these deficiencies, and more than 0.8 million human deaths every year are due to these ailments [35]. Our outcomes show increases in the Fe and Zn contents in the byproducts with respect to the edible petioles (168% in the case of Fe and 127% in the case of Zn) and an increase in Fe in the byproducts as compared to the edible leaves (15%) under conditions of a 100% N supply in non-inoculated plants not treated with Se. Moreover, the original Fe data showed a remarkable increase as a consequence of the use of a reduced N supply in fertilization, the inoculation of plants with PGPB, and the application of Se at 8 µm (23% in the petioles, 18% in the leaves, and 17% in the byproducts), with respect to the control samples.

Mn is another essential micronutrient for human health, as it is necessary for the optimization of the nervous system and immune system, and for achieving an adequate metabolism of sugars, amino acids, proteins, and lipids. In addition, traces of Mn improve the absorption of vitamins E and B1 [3,36]. There is a considerable increase when comparing the data on the Mn content in the byproducts with respect to the edible parts. In this sense, an increase of 81% was observed when comparing the Mn found in the petioles, with respect to the byproducts, and a greater increase was observed when the Mn in the leaves was compared with that in the byproducts (219%) in the samples grown under the control conditions (100% N supply, plants not inoculated with PGPB and not treated with Se). Additionally, the Mn concentration increased as result of the fertilization with a reduced dose of N and after the application of Se at 8 µm (6% in the petioles, 10% in the leaves, and 21% in the byproducts), with respect to the control samples.

### 3.3. Determination of Sugar Contents

The chromatographic profile of the sugars found in the celery tissues was composed of glucose, fructose, and sucrose (Figure 3). The total sugar contents were 18,708 mg kg^−1^ DW in the petioles, 10,907 mg kg^−1^ DW in the leaves, and 26,630 mg kg^−1^ DW in the byproducts, with glucose being the most abundant soluble sugar in all the plant parts (Figure 3). These sugar values were similar to those found in the celery variety of Lvlin Huangxinqin [37]. Nevertheless, the results found in this work were lower than those obtained for other varieties, such as Elixir, Samurai, Atlant, Primus [3], Jinnan Shiqin [38], and three celery accessions (Egypt wild type, balady, and green leaves) [39]. These findings confirm what was previously stated by Mezeyová et al. [38], who indicated that the sugar content, consisting of glucose, fructose, and sucrose, was variety-dependent in celery. In accordance with the results reported by Golubkina et al. [3], our data showed that a higher accumulation of sugars, mainly monosaccharides, was found in the petioles as compared to the leaves.

Regarding the effect of N deficiency on the accumulation of sugars, Figure 3 shows that a lower N% supply resulted in a decrease in the accumulation of sugars in the byproducts (12% at 30% N in plants not inoculated with PGPB and without the application of the Se treatment). In addition, although there were no significant differences, a trend towards a decrease in the sugars accumulated in the petioles and leaves can also be observed (Figure 3). These results are consistent with those previously obtained by Becker et al. [40], who reported a similar finding in an experiment carried out with lettuces. Individually, the sugars in the present study showed a non-uniform trend. In this sense, a notable decrease in glucose accumulation was observed as the N deficiency increased, with decreases of 10%, 29%, and 62% with respect to the control samples in the petioles, leaves, and byproducts, respectively, when only 60% N was provided, and of 23%, 39%, and 155%, with respect to the control samples, in the petioles, leaves, and byproducts, respectively, when the N supply was reduced to 30% (Figure 3). With respect to fructose, reductions of 20% and 33% in the petioles were observed when 60% and 30% N were supplied. The leaves and byproducts showed decreases of 30% and 26%, respectively, when 30% N was supplied in comparison with the values obtained in the control treatment. In contrast, the sucrose content showed a significant increase when the N supply was reduced, with an increase of 42% in the petioles with a N supply of 60%, and further increases (28% in the petioles, 31% in the leaves, and 11% in the byproducts with respect to the control samples) when only 30% N was provided (Figure 3).

These results show that sucrose synthesis was favored when the N supply decreased. In spite of the apparent lack of a relationship between the concentration of sucrose and the contents of glucose and fructose monomers [41], our data showed a greater accumulation of sucrose at the expense of glucose and fructose monomers.

The *Azotobacter salinestris* inoculation of the fertilized plants resulted in plants with a greater accumulation of sugars (61%, 20%, and 43% in the petioles, leaves, and byproducts, respectively, in those plants fertilized with a 100% N supply) (Figure 3). Our outcomes are in agreement with those of Consentino et al. [8], who reported that PGPB application significantly favored the accumulation of sugars in other green leafy vegetables, such as lettuce. Abdel Latef et al. [42] reported that the application of PGPB could lead to an increase in sugars as a PGPB mechanism that contributes to the improvement in the plant tolerance to abiotic stress.

The sugar contents increased in all the celery parts from the plants that received Se at 8 μM (43% in the petioles, 111% in the leaves, and 66% in the byproducts) in comparison with the control (Figure 3). Marked increases in the sugar contents in response to the application of Se at the appropriate concentration have been previously observed in various types of plants (quinoa, coffee, rice, cowpeas, and peanuts) [15]. Khalofah et al. [43] indicated that the increase in total sugars can be considered as a defense mechanism of plants to mitigate the abiotic stress suffered, giving rise to plants with greater resilience against abiotic stresses, such as drought.

When considering the combination of the three factors analyzed (the reduction in the N input, inoculation with PGPB, and application of Se), greater accumulations in all the sugars (59% and 68% at the 60% and 30% N supplies, respectively) in the petioles, (122% and 149% at the 60% and 30% N supplies, respectively) in the leaves and (59% and 47% at the 60% and 30% N supplies, respectively) in the byproducts, in comparison with the control plants (100% N supply, non-inoculated, and without Se), were evident. These results indicate that celery with a greater antioxidant capacity can be obtained after the application of the strategy composed of these three factors. This is because sugars play an important antioxidant role in plants under abiotic stress conditions, functioning as ROS scavengers [44].

### 3.4. Determination of Protein Contents

The protein contents in the celery tissues varied from 4.7 g 100 g^−1^ DW to 14.0 g 100 g^−1^ DW, depending on the part studied (Figure 4). These values were higher than those obtained in F1 hybrid celery plants and in the Lvlin Huangxinqin and Jinnan Shiqin celery varieties [37,45]. Gad et al. [45] reported that the fertilizer used can improve both the crop production and nutritional quality, including the protein content. Furthermore, it is necessary to point out that the rapid freezing after celery sample collection may minimize the degradation of soluble proteins [37]. In the current work, the plant’s protein contents in the different celery tissues were affected by the N level supplied in the nutrient solution, the presence of PGPB, and the application of Se at 8 µM. In this sense, when only the effect of N deficiency was studied, our results indicated that the contents of proteins decreased as the N supplied decreased (14% in the petioles, 8% in the leaves, and 7% in the byproducts at the 60% N supply, and 22% in the petioles, 15% in the leaves, and 18% in the byproducts when 30% N was supplied), as compared to the control. Although it seems that there is not a direct relationship between the soluble protein content and the dose of N applied in the fertilizing solution, in accordance with Hong et al. [22], our study showed the highest value of protein content with the highest N dose supplied. The reduction in the protein content observed can be ascribed to the possible degradation of these proteins due to improved ROS production and its concomitant oxidative damage produced by the resulting lipid peroxidation [46].

The inoculation of plants with PGPB implied the highest N accumulations (8% in the petioles, 6% in the leaves, and 15% in the byproducts) in the inoculated plants as compared to the non-inoculated ones and those fertilized with the 100% N supply (Figure 4). The positive effects of the inoculation with PGPB on increasing the content of soluble proteins have been previously studied in several crops [42]. These positive effects of PGPB on the protein content may be a direct consequence of the ability of these bacteria to fix N. However, it could also be due to the ability of these PGPB to synthesize phytohormones to solubilize inorganic phosphate, reduce the number of protein-hydrolyzing enzymes, or promote 1-aminocyclopropane-1-carboxylic acid (ACC) deaminase activity [8,42]. This enzymatic activity has been related to the ability of some bacteria to facilitate the growth of plants inoculated with these PGPB under stressful conditions. The relationship between the nutritional stress of plants caused by N deficiency and the growth potential of these PGPB emphasizes the importance of applying this technology to plants that live under stress conditions. Thus, this technology can be used to improve not only the growth of these plants, but also their nutritional quality [47].

The protein contents of the celery parts showed remarkable increases due to the application of Se (70%, 26%, and 53% in the petioles, leaves, and byproducts, respectively) with respect to the control plants. These results are in line with those of Oliveira Cunha et al. [15], who found that an adequate Se dose significantly increased the protein content in soybean. The application of Se boosted the increase in the protein content as a consequence of the inoculation of the celery plants with *Azotobacter salinestris*. Ashraf Ganjouii et al. [14] indicated that Quinoa seeds primed with Se nanoparticles and PGPB led to an increase in the protein contents, some of which may be antioxidant enzymes. Therefore, these proteins could have a direct relationship with a greater plant stress tolerance. In fact, previous studies have shown that antioxidant enzymes such as APX (ascorbate peroxidase), CAT (catalase), and POD (peroxidase) grant plants with greater resistance against abiotic stress [48].

### 3.5. Determination of Lipid Peroxidation

The lipid peroxidation results showed values from 3.3 TBARS µmol g^−1^ FW to 4.7 TBARS µmol g^−1^ FW in the plants under control conditions, depending on the celery part, with the highest value obtained in the byproducts (Figure 4). As shown in Figure 4, the highest values of lipid peroxidation in all the celery parts (petioles, leaves, and byproducts) were obtained when a lower amount of N was supplied in the fertilizer solution. This value decreased as the N provision increased. These results were consistent with those previously reported by other researchers who studied other crops, such as rice, mulberry, zea mays, morus alba, Nicotiana tabaccum, maize, Arabidopsis, and wheat [46,49]. Lipid peroxidation can be considered an indicator of oxidative damage [50].

When analyzing the values obtained for the protein content, the lipid peroxidation, and the effect of the N contribution on them, it can be clearly observed that the lipid peroxidation was greater when the protein content was lower. Thus, when the N supplied was 30%, the highest accumulation of lipid peroxidation and the lowest concentration of soluble proteins was obtained. This finding is supported by the results from a study carried out on rice by Zhang et al. [49]. The possible oxidative damage produced by lipid oxidation due to N deficiency is clearly reversed by the effects of both the application of PGPB and of Se at a concentration of 8 µm (Figure 4). In this sense, in this work, a decrease in lipid peroxidation of 24% in the petioles, as compared to the control treatment, was observed as a result of the application of the PGPB. Furthermore, when the combined effect of the PGPB with Se was studied, this reduction was reinforced, as shown by the reductions in lipid peroxidation that reached 25% in the petioles, 33% in the leaves, and 19% in the byproducts under the 100% N supply condition. Other researchers in previous works have reported that, indeed, the inoculation with PGPB, together with the application of Se, leads to a significant reduction in the concentration of malondialdehyde, an end product of lipid peroxidation, through the protection of the membrane against damages produced by different sorts of abiotic stresses, including salinity stress and nutrition stress [8,14].

There is evidence that stress triggers the biosynthesis of antioxidant compounds as a defense mechanism to fight the lipid peroxidation caused by ROS (reactive oxygen species) in plants under unfavorable conditions [51]. Considering the above, the lipid peroxidation results obtained in both the edible parts and the byproducts after the combined application of PGPB and Se to plants under N deficiency prompt us to think about a positive and efficient revalorization of celery byproducts for healthy human purposes. Furthermore, this revalorization would contribute to the circular economy.

## 4. Conclusions

The outcomes obtained in the current work revealed significant reductions in all the biomass parameters (height, petiole diameter, and dry weight), reductions in the contents of Na, P, Mn, B, sugars, and proteins in the byproducts, and increased lipid peroxidation, increased Fe in all the celery parts, and increased K in the celery byproducts as a result of a decreased N% supplied in the nutrition solution. These results were drastically changed by the inoculation of the celery plants with *Azotobacter salinestris*. In this sense, the inoculation with this kind of PGPB reversed the detrimental effects caused by the N deficiency in the fertilization treatments, resulting in a higher biomass, a higher accumulation of K in the byproducts, a build-up of sugars and proteins, and reduced concentrations of P, Cu, Mn, B, and Fe (in the petioles), Zn and K (in the byproducts), and lipid peroxidation. The application of Se at doses of 8 μM reinforced the beneficial effect obtained after inoculation with *Azotobacter salinestris*, resulting in a higher biomass and K, P, Mn, Zn, sugar, and protein accumulations and considerable decreases in the concentrations of Na, Ca, Mg, Fe, and Cu and lipid peroxidation.

In view of these results, it can be argued that the novel and proposed sustainable strategy composed of a reduced N dose, inoculation with *Azotobacter salinestris* (PGPB), and treatment with Se at 8 µm is an adequate strategy for promoting the growth of plants, and for obtaining plants with enhanced nutritional quality, in both the edible parts and the byproducts. With this improvement in nutritional quality, it is imperative to mention the significant decrease in the sodium concentration and the significant increases in K, Fe, and Zn achieved in all the celery parts, making the intake of this vegetable more recommendable as an essential ingredient in the daily diet. Furthermore, this study also reveals the great nutritional quality of the byproducts of this vegetable. For this reason, the authors of this work believe that the valorization of celery byproducts with beneficial health purposes should be taken into account. This approach could also benefit the cosmetic and nutraceutical industries.

## Figures and Tables

**Figure 1 foods-13-01437-f001:**
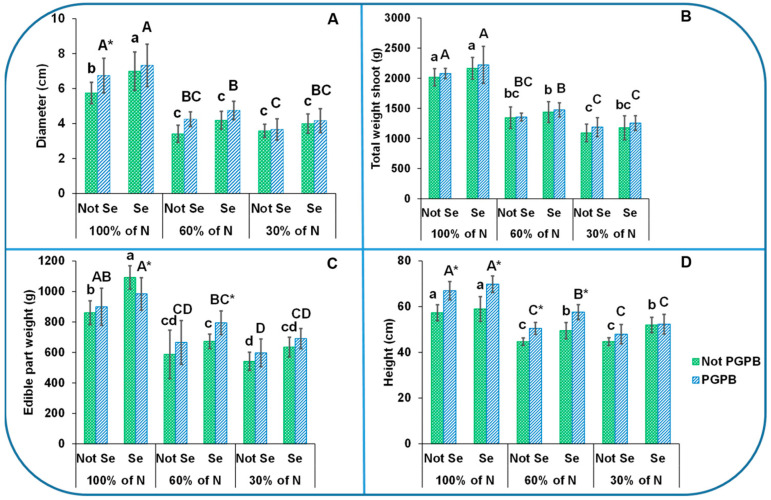
Effects of the combination of inoculation with plant growth-promoting bacteria and three different N concentrations (30%, 60%, and 100% (control) N supplies) on celery biomass parameters (diameter (**A**), total shoot weight (**B**), edible-part weight (**C**), and height (**D**)) of plants sprayed with Se at 8 µM. The data are presented as the treatment means (*n* = 5). Different lowercase letters indicate significant differences between celery plants fertilized with different N% supplies in the absence of Se. Different uppercase letters indicate significant differences between celery plants fertilized with different N% supplies sprayed with Se. Asterisks indicate significant differences between inoculated and non-inoculated plants fed with the same N dose and the same concentration of sprayed Se. Abbreviations: Not PGPB: plants not inoculated with PGPB; Not Se: plants not treated with selenium.

**Figure 2 foods-13-01437-f002:**
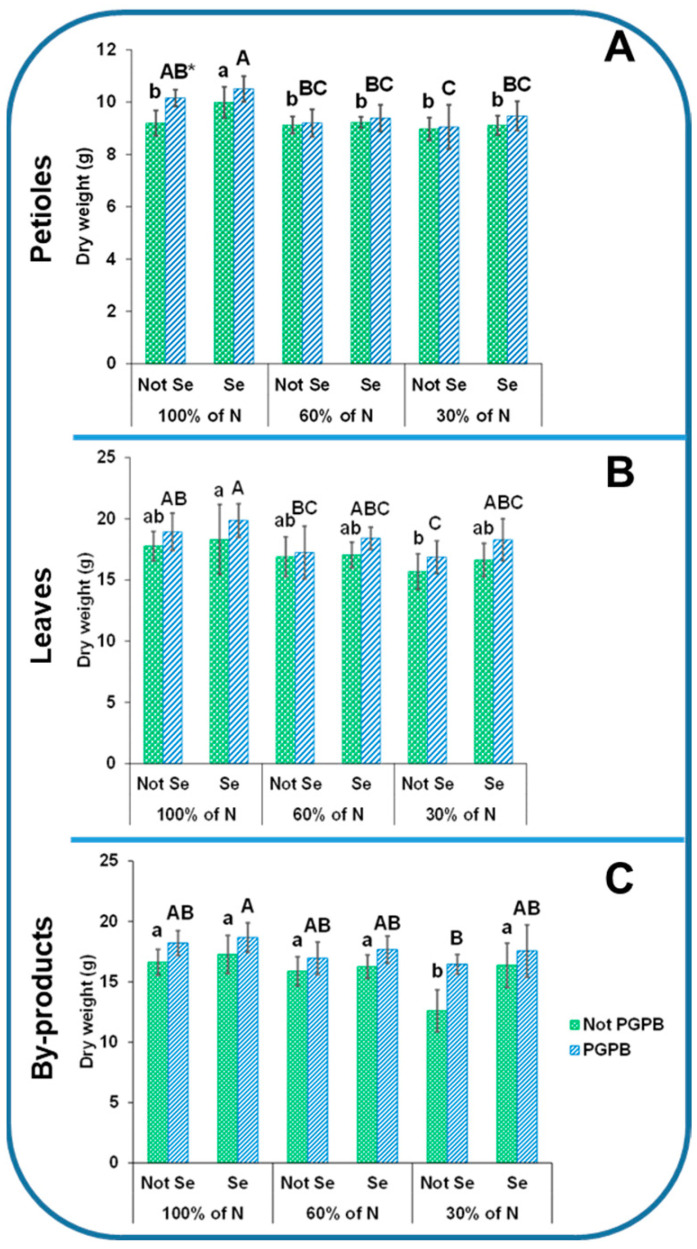
Effects of the combination of inoculation with plant growth-promoting bacteria and three different N concentrations (30%, 60%, and 100% (control) N supplies) on the dry weights of celery petioles (**A**), leaves (**B**), and byproducts (**C**) from plants sprayed with Se at 8 µM. The data are presented as the treatment means (*n* = 5). Different lowercase letters indicate significant differences between celery plants fertilized with different N% supplies in the absence of Se. Different uppercase letters indicate significant differences between celery plants fertilized with different N% supplies sprayed with Se. Asterisks indicate significant differences between inoculated and non-inoculated plants fed with the same N dose and the same concentration of sprayed Se. Abbreviations: Not PGPB: plants not inoculated with PGPB; Not Se: plants not treated with selenium.

**Figure 3 foods-13-01437-f003:**
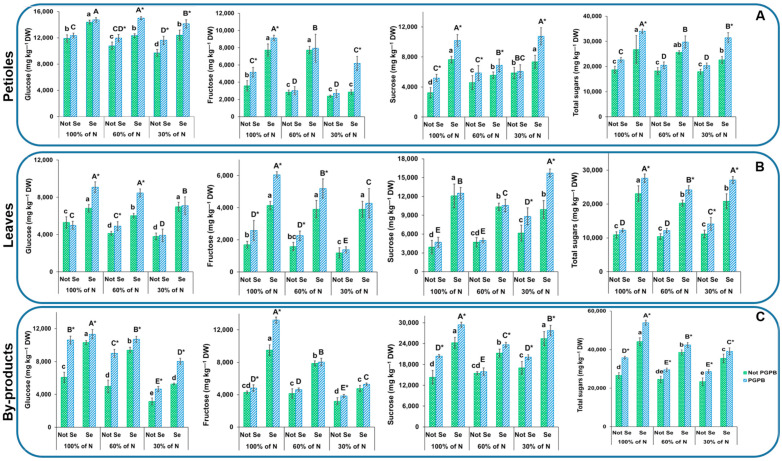
Effects of the combination of inoculation with plant growth-promoting bacteria and three different N concentrations (30%, 60%, and 100% (control) N supplies) on glucose, fructose, sucrose, and total sugar contents of celery petioles (**A**), leaves (**B**), and byproducts (**C**) from plants sprayed with Se at 8 µM. The data are presented as the treatment means (*n* = 5). Different lowercase letters indicate significant differences between celery plants fertilized with different N% supplies in the absence of Se. Different uppercase letters indicate significant differences between celery plants fertilized with different N% supplies sprayed with Se. Asterisks indicate significant differences between inoculated and non-inoculated plants fed with the same N dose and the same concentration of sprayed Se. Abbreviations: Not PGPB: plants not inoculated with PGPB; Not Se: plants not treated with selenium.

**Figure 4 foods-13-01437-f004:**
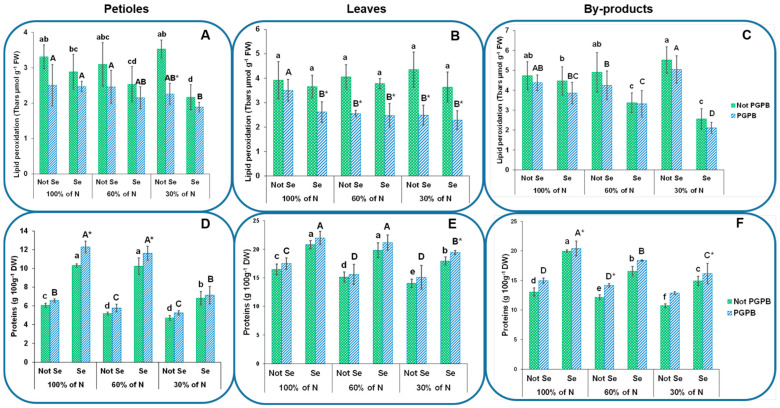
Effects of the combination of inoculation with plant growth-promoting bacteria and three different N concentrations (30%, 60%, and 100% (control) N supplies) on lipid peroxidation (**A**–**C**) and protein contents (**D**–**F**) of celery petioles, leaves, and by-products from plants sprayed with Se at 8 µM. The data are presented as the treatment means (*n* = 5). Different lowercase letters indicate significant differences between celery plants fertilized with different N% supplies in the absence of Se. Different uppercase letters indicate significant differences between celery plants fertilized with different N% supplies sprayed with Se. Asterisks indicate significant differences between inoculated and non-inoculated plants fed with the same N dose and the same concentration of sprayed Se. Abbreviations: Not PGPB: plants not inoculated with PGPB; Not Se: plants not treated with selenium.

**Table 1 foods-13-01437-t001:** Effects of the combination of inoculation with plant growth-promoting bacteria and three different N concentrations (30%, 60%, and 100% (control) N supplies) on macroelements (g kg^−1^ DW) of celery petioles, leaves, and byproducts from plants sprayed with Se at 8 µM.

Macroelements			100% N		60% N		30% N	
			Not Se	Se	Not Se	Se	Not Se	Se
Na	Pet	*Not B*	8.6 ± 0.4 ^a^	7.6 ± 0.7 ^b^	7.3 ± 0.7 ^b^	7.3 ± 0.3 ^b^	6.5 ± 0.4 ^c^	5.7 ± 0.3 ^d^
		*PGPB*	8.3 ± 0.8 ^A^	7.1 ± 0.7 ^B^	6.8 ± 0.5 ^B^	5.7 ± 0.2 ^C^*	5.5 ± 0.3 ^C^*	4.7 ± 0.3 ^D^*
	L	*Not B*	7.5 ± 0.5 ^a^	5.7 ± 0.7 ^c^	6.3 ± 0.3 ^b^	5.4 ± 0.3 ^c^	4.1 ± 0.4 ^d^	3.6 ± 0.2 ^d^
		*PGPB*	5.0 ± 0.6 ^A^*	4.7 ± 0.5 ^AB^	4.3 ± 0.5 ^B^*	3.7 ± 0.3 ^C^*	3.2 ± 0.4 ^CD^	3.1 ± 0.3 ^D^
	B–P	*Not B*	10.1 ± 0.8 ^ab^	9.8 ± 0.9 ^ab^	10.7± 0.4 ^a^	9.3 ± 0.9 ^b^	7.3 ± 0.5 ^c^	6.9 ± 0.4 ^c^
		*PGPB*	10.0 ± 1.2 ^A^	9.1 ± 1.5 ^AB^	8.7 ± 0.8 ^B^*	8.4 ± 0.3 ^B^	6.2 ± 0.3 ^C^*	5.3 ± 0.5 ^C^*
K	Pet	*Not B*	55.2 ± 4.4 ^a^	57.7 ± 2.8 ^a^	54.2 ± 2.1 ^a^	55.0 ± 2.3 ^a^	59.4 ± 6.2 ^a^	61.3 ± 4.6 ^a^
		*PGPB*	60.2 ± 5.1 ^A^	63.8 ± 4.3 ^A^*	58.0± 4.4 ^A^	61.5 ± 3.4 ^A^*	61.8 ± 5.7 ^A^	63.2 ± 3.2 ^A^
	L	*Not B*	41.8 ± 3.7 ^a^	42.3 ± 4.4 ^a^	42.6 ± 3.1 ^a^	43.0 ± 1.9 ^a^	43.1 ± 3.7 ^a^	46.3 ± 1.8 ^a^
		*PGPB*	43.8 ± 3.3 ^A^	44.6 ± 2.6 ^A^	43.8 ± 2.9 ^A^	44.0 ± 1.4 ^A^	44.6 ± 4.2 ^A^	48.2 ± 3.5 ^A^
	B–P	*Not B*	35.9 ± 2.3 ^c^	48.0 ± 3.3 ^a^	35.9 ± 3.6 ^c^	35.9 ± 3.0 ^c^	43.1 ± 1.7 ^b^	43.2 ± 2.3 ^b^
		*PGPB*	48.6 ± 3.8 ^B^*	48.8 ± 5.2 ^B^	36.2 ± 2.1 ^C^	53.3 ± 1.7 ^A^*	46.3 ± 2.6 ^B^	55.9 ± 4.2 ^A^*
Ca	Pet	*Not B*	5.4 ± 0.5 ^a^	3.8 ± 0.3 ^b^	5.9 ± 0.7 ^a^	4.3 ± 0.2 ^b^	5.9 ± 0.6 ^a^	5.6 ± 0.9 ^a^
		*PGPB*	5.0 ± 0.4 ^B^	3.6 ± 0.4 ^D^	5.3 ± 0.6 ^AB^	4.4 ± 0.3 ^C^	5.8 ± 0.5 ^A^	5.0 ± 0.5 ^B^
	L	*Not B*	12.5 ± 2.1 ^ab^	9.8 ± 0.5 ^b^	12.3 ± 2.5 ^ab^	11.5 ± 1.2 ^ab^	13.7 ± 1.1 ^a^	10.4 ± 0.9 ^ab^
		*PGPB*	12.8 ± 0.7 ^A^	8.8 ± 0.6 ^C^	13.4 ± 1.3 ^A^	10.0 ± 0.1 ^B^	12.5 ± 0.9 ^A^	10.1 ± 0.6 ^B^
	B–P	*Not B*	25.1 ± 1.8 ^ab^	23.7 ± 1.0 ^ab^	26.6 ± 1.6 ^a^	22.6 ± 0.9 ^ab^	24.3 ± 4.0 ^ab^	21.2 ± 2.4 ^b^
		*PGPB*	23.5 ± 2.9 ^A^	21.5 ± 1.8 ^A^	23.7 ± 2.5 ^A^*	18.3 ± 0.2 ^B^*	22.6 ± 1.6 ^A^	15.2 ± 1.3 ^A^*
Mg	Pet	*Not B*	2.2 ± 0.2 ^b^	1.8 ± 0.1 ^c^	2.6 ± 0.4 ^a^	1.9 ±0.1 ^c^	2.5 ± 0.4 ^a^	1.7 ± 0.1 ^c^
		*PGPB*	2.8 ± 0.2 ^A^	2.2 ± 0.1 ^B^	2.6 ±0.2 ^A^	2.4 ± 0.3 ^AB^	2.6 ± 0.1 ^A^	2.4 ± 0.4 ^AB^*
	L	*Not B*	3.4 ± 0.2 ^a^	2.8 ± 0.2 ^b^	3.4 ± 0.2 ^a^	2.7 ± 0.2 ^b^	3.4 ± 0.2 ^a^	2.4 ± 0.2 ^c^
		*PGPB*	4.0 ± 0.6 ^A^	3.3 ± 0.1 ^C^	3.8 ± 0.5 ^AB^	3.1 ± 0.3 ^C^	3.4 ± 0.3 ^BC^	2.9 ± 0.3 ^C^
	B–P	*Not B*	4.9 ± 0.3 ^a^	4.3 ± 0.3 ^b^	4.7 ± 0.2 ^a^	4.1 ± 0.4 ^b^	4.8 ± 0.2 ^a^	3.6 ± 0.2 ^c^
		*PGPB*	5.6 ± 0.3 ^A^*	4.7 ± 0.2 ^BC^	5.0 ± 0.3 ^B^	4.7 ± 0.4 ^BC^	5.0 ± 0.3 ^B^	4.6 ± 0.4 ^C^*
P	Pet	*Not B*	5.9 ± 0.1 ^b^	6.4 ± 0.2 ^a^	5.2 ± 0.1 ^c^	5.4 ± 0.3 ^c^	4.6 ± 0.3 ^d^	4.8 ± 0.2 ^a^
		*PGPB*	5.5 ± 0.3 ^B^	6.2 ± 0.4 ^A^	4.7 ± 0.1 ^D^*	5.0 ± 0.3 ^CD^	4.7 ± 0.4 ^D^	5.1 ± 0.3 ^C^
	L	*Not B*	7.7 ± 0.2 ^b^	8.6 ± 0.1 ^a^	6.8 ± 0.4 ^d^	7.2 ± 0.5 ^c^	5.4 ± 0.3 ^e^	6.7 ± 0.3 ^d^
		*PGPB*	7.2 ± 0.2 ^B^	8.7 ± 0.5 ^A^	6.3 ± 1.1 ^C^	7.2 ± 0.4 ^B^	5.8 ± 0.4 ^C^	5.9 ± 0.2 ^C^*
	B–P	*Not B*	7.4 ± 0.3 ^b^	8.6 ± 1.1 ^a^	5.3 ± 0.3 ^c^	5.6 ± 0.2 ^c^	4.6 ± 0.2 ^d^	3.9 ± 0.2 ^e^
		*PGPB*	6.6 ± 0.7 ^B^	8.2 ± 0.1 ^A^	4.2 ± 0.1 ^D^*	6.3 ± 0.2 ^B^*	5.2 ± 0.2 ^C^*	4.6 ± 0.3 ^D^*

The data are presented as the treatment means (*n* = 5). Different lowercase letters indicate significant differences between celery plants in the absence of Se. Different uppercase letters indicate significant differences between celery plants sprayed with Se. Asterisks indicate significant differences between inoculated and non-inoculated plants fed with the same N dose and the same concentration of sprayed Se. Abbreviations used: Not B: not PGPB; Not Se: not selenium; Pet: petioles; L: leaves; B–P: byproducts.

**Table 2 foods-13-01437-t002:** Effects of the combination of inoculation with plant growth-promoting bacteria and three different N concentrations (30%, 60%, and 100% (control) N supplies) on microelements (mg kg^−1^DW) of celery petioles, leaves, and byproducts from plants sprayed with Se at 8 µM.

Microelements			100% N		60% N		30% N	
			Not Se	Se	Not Se	Se	Not Se	Se
Fe	Pet	*Not B*	21.0 ± 1.9 ^b^	13.9 ± 0.9 ^d^	21.8 ± 1.3 ^b^	15.0 ± 0.8 ^d^	29.5 ± 2.5 ^a^	17.0 ± 0.8 ^c^
		*PGPB*	17.8 ± 0.5 ^E^*	20.9 ± 2.6 ^D^*	35.3 ± 2.9 ^B^*	24.3 ± 1.2 ^C^*	56.4 ± 2.9 ^A^*	25.8 ± 1.1 ^C^*
	L	*Not B*	48.9 ± 0.7 ^c^	45.0 ± 1.0 ^d^	54.0 ± 1.8 ^b^	46.4 ± 1.8 ^d^	59.8 ± 1.8 ^a^	46.4 ± 2.6 ^d^
		*PGPB*	47.8 ± 1.0 ^E^	52.6 ± 1.6 ^D^*	60.0 ± 2.3 ^B^*	54.2 ± 2.5 ^D^*	81.4 ± 1.1 ^A^*	57.7 ± 1.8 ^C^*
	B–P	*Not B*	56.1 ± 4.0 ^c^	51.0 ± 4.1 ^d^	62.8 ± 1.9 ^b^	51.0 ± 0.9 ^d^	71.9 ± 2.0 ^a^	53.5 ± 3.8 ^cd^
		*PGPB*	53.4 ± 1.7 ^D^	60.2 ± 2.4 ^BC^*	72.6 ± 4.7 ^A^*	63.7 ± 2.2 ^B^*	73.2 ± 4.0 ^A^	65.6 ± 2.1 ^B^*
Cu	Pet	*Not B*	2.2 ± 0.2 ^a^	1.6 ± 0.2 ^b^	2.1 ± 0.1 ^a^	1.7 ± 0.1 ^b^	2.1 ± 0.1 ^a^	1.3 ± 0.1 ^c^
		*PGPB*	1.2 ± 0.2 ^D^*	1.8 ± 0.1 ^C^	2.2 ± 0.2 ^A^	2.1 ± 0.1 ^AB^*	2.2 ± 0.2 ^A^	1.9 ± 0.1 ^BC^*
	L	*Not B*	2.8 ± 0.1 ^a^	2.6 ± 0.4 ^a^	2.8 ± 0.1 ^a^	2.7 ± 0.2 ^a^	2.8 ± 0.2 ^a^	2.7 ± 0.4 ^a^
		*PGPB*	2.4 ± 0.1 ^D^*	2.7 ± 0.2 ^C^	3.2 ± 0.1 ^A^	3.1 ± 0.1 ^AB^	3.0 ± 0.2 ^AB^	2.9 ± 0.1 ^B^
	B–P	*Not B*	4.5 ± 0.3 ^b^	3.1 ± 0.1 ^c^	4.8 ± 0.4 ^ab^	2.8 ± 0.1 ^c^	4.9 ± 0.3 ^a^	3.0 ± 0.2 ^c^
		*PGPB*	1.6 ± 0.2 ^C^*	2.2 ± 0.4 ^C^*	7.3 ± 1.0 ^A^*	4.1 ± 0.1 ^B^*	8.0 ± 0.6 ^A^*	4.2 ± 0.3 ^B^*
Mn	Pet	*Not B*	35.7 ± 1.0 ^bc^	44.0 ± 2.3 ^a^	33.0 ± 1.7 ^cd^	41.3 ± 2.2 ^a^	32.2 ± 2.0 ^d^	37.7 ± 0.3 ^b^
		*PGPB*	28.4 ± 1.5 ^B^*	30.7 ± 2.3 ^A^*	24.3 ± 1.6 ^C^*	29.3 ± 0.4 ^AB^*	20.0 ± 1.8 ^D^*	29.2 ± 2.4 ^AB^*
	L	*Not B*	62.8 ± 2.0 ^c^	75.1 ± 2.9 ^a^	59.7 ± 2.5 ^d^	66.1 ± 2.9 ^e^	56. 3 ± 1.9 ^e^	63.4 ± 1.9 ^bc^
		*PGPB*	47.4 ± 2.3 ^BC^*	53.0 ± 2.2 ^A^*	44.8 ± 1.8 ^C^*	49.8 ± 5.7 ^AB^*	36.9 ± 2.8 ^D^*	47.6 ± 2.1 ^BC^*
	B–P	*Not B*	114.2 ± 3.4 ^bc^	143.2 ± 13.0 ^a^	107.8 ± 7.2 ^c^	126.1 ± 2.8 ^b^	102.0 ± 7.5 ^c^	124.4 ± 5.6 ^b^
		*PGPB*	77.0 ± 2.2 ^C^*	97.6 ± 1.3 ^A^*	62.8 ± 5.2 ^D^*	86.1 ± 2.9B *	58.3 ± 3.9 ^D^*	83.8 ± 2.1 ^B^*
Zn	Pet	*Not B*	15.8 ± 0.7 ^ab^	17.3 ± 0.7 ^a^	15.4 ± 0.7 ^ab^	16.8 ± 0.9 ^a^	14.5 ± 2.4 ^b^	15.8 ± 0.5 ^ab^
		*PGPB*	14.1 ± 0.9 ^AB^	14.4 ± 0.5 ^A^*	13.9 ± 0.4 ^AB^	14.9 ± 0.4 ^A^	13.0 ± 1.6 ^B^	14.4 ± 0.9 ^A^
	L	*Not B*	37.2 ± 0.9 ^bc^	41.8 ± 1.8 ^a^	36.7 ± 2.4 ^c^	39.4 ± 3.5 ^bc^	36.2 ± 2.4 ^c^	39.2 ± 4.0 ^bc^
		*PGPB*	33.7 ± 2.5 ^A^	35.8 ± 1.7 ^A^*	33.4 ± 3.0 ^A^	34.7 ± 3.0 ^A^	33.2 ± 1.8 ^A^	34.3 ± 2.9 ^A^
	B–P	*Not B*	35.8 ± 2.5 ^ab^	39.9 ± 1.5 ^a^	35.0 ± 3.1 ^ab^	38.4 ± 5.1 ^ab^	33.9 ± 1.9 ^b^	36.6 ± 3.5 ^ab^
		*PGPB*	29.5 ± 0.8 ^AB^*	33.3 ± 1.6 ^A^*	26.8 ± 1.8 ^BC^*	31.7 ± 2.7 ^A^*	24.7 ± 2.2 ^C^*	30.7 ± 3.9 ^AB^
B	Pet	*Not B*	34.4 ± 2.8 ^b^	41.5 ± 1.6 ^a^	31.1 ± 3.6 ^b^	32.7 ± 3.2 ^b^	30.8 ± 2.0 ^b^	31.6 ± 2.8 ^b^
		*PGPB*	22.2 ± 2.0 ^BC^*	27.3 ± 1.0 ^A^*	22.1 ± 2.4 ^BC^*	23.4 ± 1.3 ^BC^*	20.8 ± 0.7 ^C^*	23.7 ± 2.0 ^B^*
	L	*Not B*	32.1 ± 2.7 ^a^	37.7 ± 5.1 ^a^	31.7 ± 1.4 ^a^	36.5 ± 3.3 ^a^	31.8 ± 4.4 ^a^	37.5 ± 3.7 ^a^
		*PGPB*	27.4 ± 1.5 ^A^*	30.5 ± 4.3 ^A^	27.7 ± 3.2 ^A^*	30.6 ± 2.1 ^A^*	26.5 ± 1.7 ^A^*	30.3 ± 3.1 ^A^*
	B–P	*Not B*	20.5 ± 1.2 ^ab^	22.5 ± 1.7 ^a^	18.1 ± 1.6 ^bc^	22.6 ± 1.5 ^a^	17.5 ± 1.4 ^c^	22.0 ± 1.5 ^a^
		*PGPB*	15.7 ± 0.8 ^ABC^*	17.4 ± 1.1 ^AB^*	15.7 ± 1.0 ^BC^*	17.4 ± 1.0 ^A^*	14.9 ± 0.7 ^C^*	17.0 ± 1.2 ^AB^*

The data are presented as the treatment means (*n* = 5). Different lowercase letters indicate significant differences between celery plants in the absence of Se. Different uppercase letters indicate significant differences between celery plants sprayed with Se. Asterisks indicate significant differences between inoculated and non-inoculated plants fed with the same N dose and the same concentration of sprayed Se. Abbreviations used: Not B: not PGPB; Not Se: not selenium; Pet: petioles; L: leaves; B–P: byproducts.

## Data Availability

The original contributions presented in the study are included in the article, further inquiries can be directed to the corresponding author.

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
