# Peer review of "Biofortification and Valorization of Celery byproducts Using Selenium and PGPB under Reduced Nitrogen Regimes"

_foods, 2024, doi:10.3390/foods13101437_

Round 1
Reviewer 1 Report
Comments and Suggestions for Authors
Research questions are well defined and within the aims and the scope of the journal. Material is accordingly defined. Methods are suitable, properly described and used in a way that is possible to replicate experiments and analyses. The investigation is performed to good technical standards. It is no ethical problem involved. Conclusions are well stated and based on the results. Discussion is sound and relevant. English style should be improved on many places.
Some specific suggestions:
The term “green wastes” should be reconsidered, if it is discussed their nutritional value.
Line 110. Instead of Azotobacter Salinestris, correct species name should be Azotobacter salinestris, and written in italic.
The same in many other lines, including 28, 551, 558.
Line 119. “nutritional quality”, this is not clear.
Line 304. “that Fe, Zn and Mn increased”, better: “that the concentration of Fe, Zn and Mn increased”.
Line 307. “it can concluded”, improve the style.
Line 309. “It is known that a high sodium intake is related to suffering from cardio vascular diseases in humans.” improve the style.
Comments on the Quality of English LanguageEnglish style should be improved on many places.
Author Response
Response to reviewer 1´s comments:
Research questions are well defined and within the aims and the scope of the journal. Material is accordingly defined. Methods are suitable, properly described and used in a way that is possible to replicate experiments and analyses. The investigation is performed to good technical standards. It is no ethical problem involved. Conclusions are well stated and based on the results. Discussion is sound and relevant. English style should be improved on many places.
Some specific suggestions:
The term “green wastes” should be reconsidered, if it is discussed their nutritional value.
Thank you so much for your piece of advice. We have changed green wastes.
Line 110. Instead of Azotobacter Salinestris, correct species name should be Azotobacter salinestris, and written in italic.
The same in many other lines, including 28, 551, 558.
Thank you so much. We have corrected and written the species name correctly.
Line 119. “nutritional quality”, this is not clear.
Thank you so much. We have changed “nutritional quality” to health-promoting effects.
Line 304. “that Fe, Zn and Mn increased”, better: “that the concentration of Fe, Zn and Mn increased”.
Thank you so much. The sentence was changed by adding “the concentration of”.
Line 307. “it can concluded”, improve the style.
Thank you so much. “It can concluded” has been changed by “it can be envisaged”.
Line 309. “It is known that a high sodium intake is related to suffering from cardio vascular diseases in humans.” improve the style.
Thank you so much. That sentence has been rewritten: There is evidence that a high sodium intake is related to an increased risk of certain chronic diseases, including heart disease in humans.

Reviewer 2 Report
Comments and Suggestions for Authors
Dear Authors,
Peer-reviewed manuscript by Jacinta Collado-González et al. is experimental work aimed at developing a new agronomic strategy that will improve plant production and at the same time increase the nutritional quality of the food produced. In terms of the following climate changes and the constantly growing population, the topic is very important and current. Additionally, the process of enriching plants, through agrotechnical methods, with elements necessary for the proper functioning of the human body is one of the main ways to prevent malnutrition.
The work is well planned and written with great care. In terms of content, there are no major objections. The introduction sufficiently covers the current state of knowledge in the field of the topic. The purpose and scope of the work were correctly defined and the adopted work methodology enabled their implementation. The authors made a competent interpretation of the obtained results, which were discussed based on the current literature on the subject.
The reviewer's greatest reservation is the graphical presentation of the research results. The charts are too small and illegible. For this reason, there is a need to reword them.
The second caveat is related to the description of the results. The authors constantly provide the value of a specific parameter, e.g. for leaves, without specifying which part of the experiment this result comes from, e.g. from the control (lines 200-202) or they write that a given parameter increases, but they do not specify in comparison to what (e.g. lines 357-360 ). This requires necessary corrections throughout the manuscript.
Below is a list of minor changes and additions that the reviewer believes will improve the manuscript:
- I suggest changing the title so that it doesn't sound like a sentence
- the spelling of Azotobacter salinestris is questionable. If Azotobacter salinestris is a species, both words should be written in italics, the first with a capital letter and the second with a lowercase (Azotobacter salinestris), and if it is a serotype, the first part should be written in capital letters and italics, while the second part should be written in capital letters normally (without italics) (Azotobacter Salinestris)
- keywords - I suggest leaving PGPB alone without "and sustainable strategy" but expanding this abbreviation; leave valorization alone (delete "of celery green wastes")
Lines 49-51 – I suggest adding what the harmful effects of nitrates are for plants and humans
Lines 84-85 – by-products are not part of celery (reword)
Line 106 – a dot is missing in the magnesium sulfate hydrate formula
Line 150 – the word "cations" in lower case
Lines 156, 168, 174, 186, 188, 452, 453, 499, 500 – in the unit of cation content -1 should be in the superscript
Lines 159-160 – how exactly was the extract prepared?
Lines 160, 162, 171, 183 – without spaces between the number and the degrees symbol (e.g. 4oC)
Line 181 – space between value and unit
Line 184 – please provide equipment
Line 230 – the caption says "not Se" while the charts say "no Se" - please standardize throughout the manuscript
Line 231 – "The dry matter..." add (figure 2)
Lines 282-284 - please clarify this sentence with information about what macro and microelements in what parts of the celery increased
Line 298 – delete "Regarding Se"
Line 309 - the sentence "It is known..." suggests moving to the next paragraph starting with the sentence "Our results..."
Table 1 – in line 156 a different unit is given than in the table caption – please check in what units the values ​​in the table are given
Table 2 – the Fe content value for Pet Not B at 60% N is incorrect
Lines 358-360, 370 and 372 (and elsewhere in the manuscript) – are these values given in relation to the control sample?
Lines 378-381 – total sugar content is given for control? This is not clear from the charts, especially since the chart gives a different unit than the one in the text
Line 496 – please explain enzyme abbreviations
Lines 499-500 – are the given values correct? For which part of the experiment are they given? If for control purposes, the charts show something different.
Line 545 – without a dot
Line 547 – the word "content" is missing Na, P, Mn….
Best regards,
Reviewer
Author Response
Response to reviewer 2´s comments:
Peer-reviewed manuscript by Jacinta Collado-González et al. is experimental work aimed at developing a new agronomic strategy that will improve plant production and at the same time increase the nutritional quality of the food produced. In terms of the following climate changes and the constantly growing population, the topic is very important and current. Additionally, the process of enriching plants, through agrotechnical methods, with elements necessary for the proper functioning of the human body is one of the main ways to prevent malnutrition.
The work is well planned and written with great care. In terms of content, there are no major objections. The introduction sufficiently covers the current state of knowledge in the field of the topic. The purpose and scope of the work were correctly defined and the adopted work methodology enabled their implementation. The authors made a competent interpretation of the obtained results, which were discussed based on the current literature on the subject.
The reviewer's greatest reservation is the graphical presentation of the research results. The charts are too small and illegible. For this reason, there is a need to reword them.
Thank you so much. We have improved the figures resolution.
The second caveat is related to the description of the results. The authors constantly provide the value of a specific parameter, e.g. for leaves, without specifying which part of the experiment this result comes from, e.g. from the control (lines 200-202) or they write that a given parameter increases, but they do not specify in comparison to what (e.g. lines 357-360 ). This requires necessary corrections throughout the manuscript.
Thank you so much. That information has been added. Please, see lines: 238-240; 243-244; 357- 360; 371- 375; 393-394; 401-403; 407-411; 422- 423; 441- 445; 490-491; 523-524.
Below is a list of minor changes and additions that the reviewer believes will improve the manuscript:
- I suggest changing the title so that it doesn't sound like a sentence
Thank you so much. We have changed it: Biofortification and valorization of celery by-products using selenium and PGPB under reduced N regimes.
- the spelling of Azotobacter salinestris is questionable. If Azotobacter salinestris is a species, both words should be written in italics, the first with a capital letter and the second with a lowercase (Azotobacter salinestris), and if it is a serotype, the first part should be written in capital letters and italics, while the second part should be written in capital letters normally (without italics) (Azotobacter Salinestris)
Thank you so much. We have corrected and written the species name correctly.
- keywords - I suggest leaving PGPB alone without "and sustainable strategy" but expanding this abbreviation; leave valorization alone (delete "of celery green wastes")
Thank you so much. We have followed your suggestions.
Lines 49-51 – I suggest adding what the harmful effects of nitrates are for plants and humans
Thank you so much. We have added that information to the main text: “One of the main concerns is the relationship between the intake of nitrates in the diet and the incidence of suffering from methemoglobinemia, which causes blue baby syndrome. This disease is common in babies who have ingested a high concentration of nitrates through food. Another important worry is the conversion of nitrates into highly carcinogenic compounds called nitrosamines [7]”.
Lines 84-85 – by-products are not part of celery (reword)
Thank you so much. We have reworded that sentence.
Line 106 – a dot is missing in the magnesium sulfate hydrate formula
Thank you so much. The dot has been added: MgSO4 ·7H2O.
Line 150 – the word "cations" in lower case
Thank you so much. It has been corrected.
Lines 156, 168, 174, 186, 188, 452, 453, 499, 500 – in the unit of cation content -1 should be in the superscript
Thank you so much. The unit has been corrected.
Lines 159-160 – how exactly was the extract prepared?
Thank you so much. We have written in the main text how we prepared the extracts. Please, see lines 163- 166.
Lines 160, 162, 171, 183 – without spaces between the number and the degrees symbol (e.g. 4oC)
Thank you so much. This spaces have been removed.
Line 181 – space between value and unit
Thank you so much. It has been corrected.
Line 184 – please provide equipment
Thank you so much. In the main text we have has been added the UV–visible spectrophotometer (Shimadzu UV-1800 model with the CPS-240 cell holder, Shimadzu Europa GmbH, Duisburg, Germany). Please see lines 187-190.
Line 230 – the caption says "not Se" while the charts say "no Se" - please standardize throughout the manuscript
Thank you so much. We have standardized the figures and manuscript.
Line 231 – "The dry matter..." add (figure 2)
Thank you so much. We have added (Figure 2) for the purpose of clarification.
Lines 282-284 - please clarify this sentence with information about what macro and microelements in what parts of the celery increased
Thank you so much. Dear reviewer we are think that that information is already included in the text “The most accumulated mineral elements in celery were potassium (K), calcium (Ca), iron (Fe), and manganese (Mn). The highest mineral content was noted in the by-products, followed by leaves”.
Line 298 – delete "Regarding Se"
Thank you so much. “Regarding Se” has been removed.
Line 309 - the sentence "It is known..." suggests moving to the next paragraph starting with the sentence "Our results..."
Thank you so much. “It is known” has been changed to “There is evidence that”.
Table 1 – in line 156 a different unit is given than in the table caption – please check in what units the values in the table are given
Thank you so much. The units have been corrected: “The cation contents of macronutrients and micronutrients were expressed as g kg-1 DW and mg kg-1 DW, respectively”. Please, see lines 160-162.
Table 2 – the Fe content value for Pet Not B at 60% N is incorrect
Thank you so much. That content has been corrected.
Lines 358-360, 370 and 372 (and elsewhere in the manuscript) – are these values given in relation to the control sample?
Thank you so much. We have clarified that those values were compared with control plants. Please, see lines 238-240; 243-244; 357- 360; 371- 375; 393-394; 401-403; 407-411; 422- 423; 441- 445; 490-491; 523-524.
Lines 378-381 – total sugar content is given for control? This is not clear from the charts, especially since the chart gives a different unit than the one in the text
Thank you so much. Yes, the total sugar content mentioned was from plants not inoculated with PGPB and without the application of the Se treatment. This has been added for clarification in the text. Please, see line 383- 389. The units in figure 3 have been corrected.
Line 496 – please explain enzyme abbreviations
Thank you so much. The enzyme abbreviations have been explained: APX (ascorbate peroxidase), CAT (catalase), and POD (peroxidase).
Lines 499-500 – are the given values correct? For which part of the experiment are they given? If for control purposes, the charts show something different.
Thank you so much. We regret the confusion. The sentence was poorly expressed. Now it has been corrected. Please, see lines 507-509.
Line 545 – without a dot
Thank you so much. This has been corrected.
Line 547 – the word "content" is missing Na, P, Mn….
Thank you so much. It has been solved. Please, see line 555.
Best regards,
Reviewer
